



# Overview of NASA's MODIS and VIIRS Snow-Cover Earth System Data Records

George A. Riggs[1], Dorothy K. Hall[2,3] and Miguel O. Román[2]

[1]SSAI, Inc., 10210 Greenbelt Road, Suite 600, Lanham, MD 20706
[2]Terrestrial Information Laboratory, Code 619, NASA/GSFC, Greenbelt, MD 20771
[3]Michigan State University, Department of Geography, East Lansing, MI 48824

*Correspondence to*: George Riggs (george.a.riggs@nasa.gov)

**Abstract.** Knowledge of the distribution, extent, duration and timing of snowmelt is critical for characterizing the Earth's climate system and its changes. As a result, snow cover is one of the Global Climate Observing System (GCOS) essential
climate variables (ECVs). Consistent, long term datasets of snow cover are needed to study interannual variability and snow climatology. The NASA snow-cover datasets generated from the Moderate Resolution Imaging Spectroradiometer (MODIS) on the Terra and Aqua spacecraft and the Suomi National Polar-orbiting Partnership (Suomi-NPP) Visible Infrared Imaging Radiometer Suite (VIIRS) are NASA Earth System Data Records (ESDR). The objective of the snow-cover detection algorithms is to optimize the accuracy of mapping snow-cover extent (SCE) and to minimize snow-cover detection
errors of omission and commission using automated, globally-applied algorithms to produce SCE data products. Advancements in snow-cover mapping have been made with each of the four major reprocessings of the MODIS data record which extends from 2000 to the present. MODIS Collection 6 (C6) and VIIRS Collection 1 (C1) represent the state-of-the-art global snow-cover mapping algorithms and products for NASA Earth science. There were many revisions made in the C6 algorithms which improved snow-cover detection accuracy and information content of the data products. These
improvements have also been incorporated into the NASA VIIRS snow cover algorithms for C1. Both information content and usability were improved by including the Normalized Snow Difference Index (NDSI) and a quality assurance (QA) data array of algorithm processing flags in the data product, along with the SCE map. The increased data content allows flexibility in using the datasets for specific regions and end-user applications. Though there are important differences between the MODIS and VIIRS instruments (e.g., the VIIRS 375 m native resolution compared to MODIS 500 m), the
snow-detection algorithms and data products are designed to be as similar as possible so that the 16+ year MODIS ESDR of global SCE can be extended into the future with the S-NPP VIIRS snow products and with products from future Joint Polar Satellite System (JPSS) platforms. These NASA datasets are archived and accessible through the NASA Distributed Active Archive Center (DAAC) at the National Snow and Ice Data Center (NSIDC) in Boulder, Colorado.

DOIs of the referenced datasets:

MODIS Collection 5

doi: http://dx.doi.org/10.5067/ACYTYZB9BEOS

doi: http://dx.doi.org/10.5067/R90VAMI75N22

doi: http://dx.doi.org/10.5067/63NQASRDPDB0

doi: http://dx.doi.org/10.5067/ZFAEMQGSR4XD

doi: http://dx.doi.org/10.5067/EI5HGLM2NNHN

doi: http://dx.doi.org/10.5067/EW53FPU9NAS6

MODIS Collection 6

doi: http://dx.doi.org/10.5067/MODIS/MOD10_L2.006

doi: http://dx.doi.org/10.5067/MODIS/MYD10_L2.006

doi: http://dx.doi.org/10.5067/MODIS/MOD10A1.006

doi: http://dx.doi.org/10.5067/MODIS/MYD10A1.006

doi: http://dx.doi.org/10.5067/MODIS/MOD10C1.006

doi: http://dx.doi.org/10.5067/MODIS/MYD10C1.006

VIIRS Collection 1

doi:10.5067/VIIRS/VNP10.001

## 1 Introduction

NASA's Moderate Resolution Imaging Spectroradiometer (MODIS) snow-cover data products have been available since early 2000 following the December 18, 1999, launch of the Terra satellite. A second MODIS instrument was launched on May 4, 2002, on the Aqua satellite. Both the Terra and Aqua MODIS instruments are still functioning. There are now more

than 16 years of daily, 8-day and monthly snow-cover extent (SCE) data products available. MODIS snow-cover products have been improved over time, with each of the four rounds of reprocessing of the data record. The MODIS snow-cover products have been used with increasing sophistication over time, for a wide variety of studies by the snow, hydrology and climate communities. On October 28, 2011, the Visible Infrared Imaging Radiometer Suite (VIIRS) was launched on board the Suomi National Polar-orbiting Partnership (S-NPP) satellite. The MODIS and VIIRS instruments are similar in design

and both are well-suited for mapping snow cover. The NASA VIIRS snow-cover Collection 1 (C1) algorithms and data products are also very similar to the current MODIS Collection 6 (C6) algorithms and products.

The capability to map snow cover accurately from space is reflected in the satellite supplement of the Global Climate Observing System's (GCOS) terrestrial essential climate variables (ECVs) Implementation Plan (Bojinski et al., 2014). In

particular, knowledge of the distribution, extent, duration of snow and snowmelt timing has been noted by GCOS as critical for characterizing the Earth's climate system and its changes. Seasonal snow cover can have a very high surface albedo, reflecting 80% or more of the incident solar radiation back to space. In some years, approximately one third of the Earth's

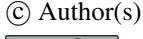


total land area can be snow covered during the Northern Hemisphere winter. Changes in the timing, density and thickness of snow cover influence the exchange of heat between the air and underlying ground. In addition, especially in mountainous areas such as in the western United States, snowmelt is a critical water resource, and is also used to generate hydropower in many areas of the world.

The overarching objective of mapping snow cover with the MODIS and VIIRS instruments is to develop a consistent long-term snow-cover dataset based on a globally-applicable algorithm that is able to detect snow cover under a wide variety of solar viewing angles in different land covers and seasonal conditions. The specific objective of the algorithms used to generate the MODIS snow-cover products in C6 and the NASA VIIRS C1 is to optimize the accuracy of mapping snow-
cover extent (SCE) on the global scale and minimize snow-cover detection errors of omission and commission.

There have been periodic reprocessings of the MODIS snow products when advances in algorithms or sensor calibration knowledge have shown to significantly improve product quality or utility. Improvements in the MODIS snow products have been incremental since data processing began in 2000. The C6 MODIS SCE product suite is available from the National
Snow and Ice Data Center (NSIDC) Distributed Active Archive Center (DAAC). Collection 5 (C5) consists of a mature suite of MODIS land and cryospheric science algorithms that include SCE maps with an accuracy range of up to 95% under clear skies (Tong et al., 2009; Gao et al., 2010a; Gao et al., 2010b; Arsenault et al., 2012; Crawford, 2015; Gascoin et al., 2015; Marchane et al., 2015; Mir et al., 2015; Hawotte et al., 2016). The Terra and Aqua MODIS snow-cover algorithms and data products in C6 have been significantly revised and data content increased compared to C5 (Riggs et al., 2016a;
Riggs et al., 2016b; Riggs et al., 2016c). Information on the MODIS and VIIRS snow cover algorithms and datasets is posted on the MODIS and VIIRS Snow and Ice Global Mapping Project website (modis-snow-ice.gsfc.nasa.gov). Descriptions and discussion of the algorithms are also given in the MODIS C6 and NASA VIIRS C1 Algorithm Theoretical Basis Documents (ATBD) and user guides (Riggs et al., 2016a; Riggs et al., 2016b; Riggs et al., 2016c). Revisions in the snow-mapping algorithms from the latest MODIS C6 are described in detail in the MODIS Collection 6 Snow Products User
Guide (Riggs et al., 2016a).

In this paper we describe the MODIS C6 and NASA VIIRS C1 snow-cover products, as well as the major improvements and enhancements in the MODIS snow-cover algorithm as provided in the currently-available snow products in C6 and NASA VIIRS C1 data collections. First we discuss the normalized difference snow index (NDSI) as the basis for the MODIS and
VIIRS snow detection algorithm used to generate the products.



## 2 Background

Snow cover was first observed and mapped from space from the Television Infrared Observation Satellite-1 (TIROS-1) satellite in 1960. In 1966 the National Oceanic and Atmospheric Administration (NOAA) started to produce weekly SCE maps of the Northern Hemisphere using a variety of satellites and ground measurements (Matson et al., 1986). NOAA's

National Ice Center (NIC) now produces daily maps in the Interactive Multisensor Snow and Ice Mapping System (IMS) (Ramsay, 1998; Helfrich et al., 2007) at a spatial resolution of up to 1 km. Using NOAA's 50-year snow-cover record, the Rutgers University Global Snow Lab (RUGSL) produces and maintains a climate-data record (CDR) of Northern Hemisphere snow cover (http://climate.rutgers.edu/snowcover/) (Frei and Robinson, 1999; Robinson, 2013).

In 2000, the MODIS snow-cover maps from the Terra satellite became the first fully-automated, global snow-cover maps produced from satellite data. Developed at NASA's Goddard Space Flight Center, a suite of MODIS snow-cover map products has been produced that continues as Collection 6, providing data free to users worldwide (Hall et al., 2002; Riggs et al., 2016a), Data products are orderable from the NSIDC DAAC (https://nsidc.org/data/modis/index.html). The early MODIS snow-mapping algorithms were developed based on heritage work using Landsat Thematic Mapper (e.g., Dozier

and Marks, 1987) and MODIS Airborne Simulator (MAS) data (Hall et al., 1995).

Many diverse studies have been conducted in which the MODIS snow-cover products have been used successfully. The reader is referred to two bibliographies of publications that report results in which the snow products were used: http://modis-snow-ice.gsfc.nasa.gov/?c=publications and http://nsidc.org/data/modis/research.html. Early studies tended to

focus on validation of the MODIS snow-cover products (e.g., Klein and Barnett, 2003; Ault et al., 2006; Parajka and Blöeschl, 2006; Parajka and Blöeschl, 2012). Several issues with the snow-mapping algorithms and data products that were identified in some of those studies have been resolved or improved on in C6. The MODIS snow-cover maps are also well-suited for modeling studies such as for validation of model outputs (e.g., Rodell and Houser, 2004; Déry et al., 2005). The daily MODIS SCE product has also been used to accurately track seasonal snow cover on the landscape, as demonstrated for

a small region of the Appalachian Mountains in West Virginia (Riggs and Hall, 2014). Increasingly, researchers are using the 16+ year MODIS time series to study changes and even to look for trends in regional SCE (Pu et al., 2007; Hall et al., 2012; Dietz et al., 2012; Dietz et al., 2013) especially when combined with datasets such as Landsat having longer time series (e.g., Hall et al., 2015). As the MODIS records get longer, time series data become still more valuable.

There is ongoing work focused on merging the MODIS and (the shorter) VIIRS snow-cover records to enable development of an Earth System Data Record (ESDR) that will eventually span several decades (Justice et al., 2013). The VIIRS on the Joint Polar Satellite System-1 (JPSS-1) is scheduled for launch in 2017, with follow-on JPSS satellites planned.



The MODIS SCE products are produced as a sequence of products starting with the Level-2 swath product, M*D10_L2, which is then composited and projected to make the daily tile product M*D10A1, which is then composited to generate the daily global snow-cover product, M*D10C1. (Terra (MOD) and Aqua (MYD) products are indicated by M*D.) The snow-cover detection algorithm is applied at the swath level; the higher level products are produced with compositing, projection

and gridding algorithms. The VIIRS snow-cover products are produced in a similar sequence. Brief descriptions and examples of the higher level products are presented in Sect. 4 and 5, and detailed descriptions are given in the snow-cover data products user guides (Riggs et al., 2016a; Riggs et al., 2016b). The theory and application of the snow detection algorithm, discussed in Sect. 3 is the foundation of all the standard NASA MODIS and VIRS snow-cover data products.

## 3 Normalized Difference Snow Index (NDSI) as the standard for global snow mapping

The primary characteristics used to detect snow cover on the landscape using satellite sensors with spectral bands in the visible and infrared wavelength regions are high visible (VIS) reflectance across the 0.3 – 1.0 μm wavelength region, and low shortwave-infrared (SWIR) (0.9 – 1.7 μm) reflectance. High VIS reflectance and low SWIR reflectance is an intrinsic (but not a unique) optical characteristic of snow, however the relative magnitude of VIS and SWIR reflectance difference over snow can vary with snow conditions and illumination. The difference in reflectance (VIS – SWIR) is greatest under

sunlit conditions of pure snow and high solar elevation angle and clear atmosphere. Snow in other situations, such as under low illumination or under a vegetation canopy, may exhibit a relatively low reflectance difference between VIS and SWIR, and NDSI values may range from high to low.

These characteristics have been examined in numerous snow reflectance curves reported in the literature. Snow reflectance

curves have been obtained in the field from spectrometers, e.g. Bowker et al. (1985) and Satterwhite et al. (2003), showing sunlit snow surfaces and shaded snow, and snow of different ages or with contamination of the surface (Singh et al., 2010; Zhang and Zhou, 2011) or, using airborne hyperspectral sensors (Punia and Dhankar, 2014). The curves demonstrate this intrinsic property of snow having high VIS and low SWIR reflectance under many different physical snow or illumination/viewing conditions.

VIS and SWIR reflectance ratios have been used to detect snow cover since at least the mid 1970s. The first documented use of ratios of VIS and NIR or SWIR was to separate snow and clouds by Valovcin (1976 & 1978) and Kyle et al. (1978), with considerable more work done by Bunting and d'Entremont (1982) who developed an automated algorithm to discriminate snow cover from clouds for global cloud analysis. In the 1980s, research focused on using the VIS and NIR

ratio technique to refine algorithms for snow cover detection with notable contributions by Crane and Anderson (1984) and Dozier (1989), with the refinement of using a normalized difference of VIS and NIR at regional scales by Rosenthal and Dozier (1996), and by Riggs et al. (1993), Hall et al. (1995), Hall et al. (2002) and Hall and Riggs (2007) for global snow-





cover mapping. In those techniques, SCE was determined by setting a threshold value of the ratio or normalized difference to make a binary map of SCA. A history of the use of VIS to SWIR ratios and the NDSI is given in Hall and Riggs (2011).

The MODIS C6 and VIIRS C1 snow-detection algorithm is based on the NDSI with data screens applied to alleviate snow
detection commission errors and to flag pixels for which snow detection is uncertain. The NDSI is an index of the presence of snow in a pixel and is a more accurate description of the snow detection as compared to the Fractional Snow Cover (FSC) output that was provided in the MODIS C5 products.

In the MODIS and VIIRS snow-mapping algorithms, NDSI is defined by the difference in VIS and SWIR reflectance in the
following channels:

NDSI for MODIS = (band 4 - band 6) / (band 4 + band 6),          (1)

where band 4 is centered at 0.56 μm and band 6 is centered at 1.64 μm.

NDSI for VIIRS = (band I1 – I3) / (band I1 + I3),          (2)

where band I1 is centered at 0.64 μm, and band I3 is centered at 1.61 μm.

Snow cover has an NDSI > 0.0. However other features e.g. salt pans, and cloud contaminated pixels at the edges of cloud, and features with very low visible reflectance can have NDSI > 0.0, and thus be erroneously detected as snow, which results in a snow commission error (detecting snow where there is no snow). Data screens for atypical snow reflectance characteristic are applied to alleviate snow commission errors and increase the accuracy of snow mapping.

In general, the greater the VIS-SWIR difference, the higher the NDSI. As the VIS reflectance increases, the difference between VIS and SWIR reflectance increases, assuming a near-constant SWIR reflectance, thus increasing the NDSI to a maximum NDSI of 0.81 as shown for specific conditions in Fig. 1. This greater difference increases the certainty of snow-cover detection. Conversely, with a lower difference between VIS and SWIR reflectance the certainty of snow cover
detection decreases, although the NDSI may have a relatively high value.

Mishra et al. (2009) investigated the range of NDSI values of snow in the Himalayan region relative to subpixel snow modeling and found that NDSI can range from 0.04 to 0.92 as the amount of snow cover increases. The NDSI can provide more information than a binary decision on SCE (e.g., Kolberg and Gottschalk, 2010; Dobreva and Klein, 2011). This is in
part because the use of an NDSI threshold to create a binary snow-cover decision for each pixel ignores the ability of the NDSI to map snow at lower values. If an NDSI threshold for snow cover is set at 0.4 (see Fig. 1), then all snow cover with



lower values will be excluded from the snow map, which could eliminate a significant amount of snow (e.g., Jain et al., 2008; Hassan et al., 2012; Lin et al., 2012).

NDSI values on a snow-covered landscape range from 0.0 to 1.0, with the highest NDSI values found in areas where snow is

a dominant feature, e.g. snow-covered plains and lower NDSI values are observed for where snow is mixed with other features, e.g. snow-covered forest in Fig. 2. The NDSI is affected by landscape features and viewing conditions, thus a positive NDSI value can represent a wide range of snow-cover conditions as shown for MODIS in Fig. 2 and for VIIRS in Fig. 3.

Scatter plots of NDSI versus VIS-SWIR in Fig. 4 and 5 for the scenes shown in Fig. 2 and 3, respectively, demonstrate the spread of NDSI values. Distribution of NDSI values for each scene in Fig. 2 and 3 is shown in the histogram plots (Fig. 4 and 5, respectively). Note the "tail" of lower range NDSI values for both scenes. Lower values correspond to the snow-covered boreal forests, and more challenging viewing conditions, while higher NDSI values correspond to the snow-covered plains and mountains. In general, the NDSI values for snow cover increase as the difference in VIS and SWIR reflectance

increases (see Fig. 1). That range and increase in NDSI values depends on the specified amount of constant SWIR reflectance. When SWIR reflectance is very low, as for nearly pure snow, NDSI values rise rapidly, but in snow-covered landscapes where for example the boreal forests canopy increases the SWIR reflectance, the NDSI values tend to be lower and increase slowly for increasing VIS-SWIR differences.

**3.1 Accuracy, Uncertainty and Error**

Accuracy applies to the mapping of snow covered area on the landscape. Uncertainty and error applies to detection of snow cover at the pixel level. Snow cover has an NDSI > 0 but not all features with an NDSI > 0 are snow cover. Non-snow features with NDSI >0 are a source of snow commission error in the algorithm. To increase accuracy, reduce commission errors and flag uncertain snow cover detection a series of data screens is applied in the algorithm. The data screens are based

on reflectance characteristics of snow and one is based on surface temperature. These data screens were revised in MODIS C6 based on evaluation of C5 products and studies of the use of C5 products reported in the literature. The accuracy of the heritage MODIS C5 daily snow-cover product has been reported in a diversity of landscapes, with different methods of comparison (e.g. Hall and Riggs, 2007; Jain et al., 2008; Huang et al., 2011; Arsenault et al., 2012; Rittger et al., 2013; Marchane et al., 2015). The accuracy of the MODIS C6 products is similar to, or better than MODIS C5, with a notable

improvement in reduction of snow omission error in Northern Hemisphere mountains during springtime as described in Sect. 3.1.1. The C6 products were released in July 2016 and therefore independent evaluations of their accuracy have not yet been published to our knowledge.



Uncertainty and errors of snow detection at the pixel level are affected by viewing conditions, land cover, amount of snow, amount of VIS and SWIR reflectance from the surface, and cloud masking. A series of data screens is applied in the algorithm to snow detections to alleviate snow errors and flag uncertain snow detections based on reflectance features that

are atypical of snow. Results of these data screens are set as bit flags in a Quality Assurance (QA) dataset in the MODIS C6 and VIIRS C1 snow products (Level-2 and daily gridded products). The data screens are described in the MODIS Snow Products User Guide (Riggs et al., 2016a) and VIIRS snow cover ATBD (Riggs et al., 2016c) and user guide (Riggs et al., 2016b) and are not repeated here with the exception of a description the surface temperature screen which caused notable snow-cover omission errors in C5.

**3.1.1 Surface temperature screen**

The surface temperature screen had a detrimental effect in C5 and has been revised for MODIS C6 and VIIRS C1. The surface temperature screen in C5 caused the reversal of snow-cover detection to "no snow" on some mountain ranges during spring and summer seasons. On mountains, mixed pixels with snow, rocks and/or vegetation can have an estimated surface temperature greater than 0° C because of the contribution of non-snow features to the surface temperature of the pixel. In

C5, the surface temperature screen was set at 283K for the purpose of reducing snow commission errors of "non-snow" features that are spectrally similar to snow. The C5 algorithm reversed snow detection to "snow-free land" if the MODIS surface temperature was above the 283K threshold. However, it was common in spring and early summer on some mountain ranges for mixed pixels that included snow to have surface temperatures above 283K and thus a great deal of spring or summer snow on mountains was not mapped, resulting in significant snow commission errors on some mountain

ranges in spring and summer in C5.

Rittger et al. (2013) compared MODSCAG FSC to the MOD10A1 C5 binary and FSC snow cover products. They showed that snow cover in the seasons of autumn, spring and summer on some mountain ranges was not mapped in the MOD10A1 C5 FSC data product (Rittger et al., 2013, Fig. 6). The reason for the missing snow was the application of the surface

temperature screen in the C5 snow-cover algorithm. This was a known problem that was identified and fixed in C6. The effect of the surface temperature causing "missing" snow has been highlighted on the MODIS snow cover project website http://modis-snow-ice.gsfc.nasa.gov/?c=collection6 for the Sierra Nevada Range, has been posted there for several years. Application of the surface temperature screen was revised in C6. It is linked with surface height and does not reverse snow detection at heights above 1300 m and a QA flag is set in a QA data layer which is included in the C6 MOD10A1 product.

As a result, snow cover is not missed in MOD10A1 C6 for the situations report by Rittger et al. (2013) in their Fig. 6 for the Sierra Nevada and Upper Rio Grande. The MOD10A1 C6 NDSI snow-cover maps for those regions are shown in Fig. 6. A true color image from MOD09GA C6 along with the MOD10A1 C5 FSC and MOD10A1 C6 NDSI snow cover are shown in Fig. 6. The regions correspond in coverage to those shown in Rittger et al. (2013) Fig. 6 (a greater areal extent of the regions



is shown here). The bottom line is that snow cover is not missing in MOD10A1 C6 NDSI snow cover where it was missing in MOD10A1 C5.

Linking the surface temperature screen with surface height allows the screen to be applied at lower elevations where it is most effective (< 1300 m). This screen combination is also applied and set as quality flags at higher elevations but does not cause a reversal of a snow detection decision. Surface temperature is estimated as the brightness temperature of VIIRS band I5 and the surface temperature threshold is set at 281K which is too warm for snow. If a snow detection is at a height < 1300 m and too warm, it is reversed to "not snow" and a QA bit flag is set. If snow detection is at ≥ 1300 m and too warm (above 281K), the QA bit flag indicates warm snow detection.

The effectiveness of the surface temperature screen linked with surface height is shown in Fig. 7, where snow commission errors on cloud-shadowed forests in Brazil are changed to "not snow."

### 3.1.2 Solar illumination

The highest potential accuracy of the snow detection algorithm occurs when solar zenith angles are ≤70°. At solar zenith angles in the range of 70° to 85°, uncertainty in the retrieval of land biogeophysical parameters in general, and snow detection in particular, increases because of the low illumination (Schaaf et al., 2011). In MODIS/VIIRS snow processing, night is defined by pixels with a solar zenith angle ≥ 85°. Low illumination in the solar zenith angle range of 70° to 85° affects much of the Northern Hemisphere in the winter, making it a challenge to detect snow. Therefore a High-Solar-Zenith-Angle-Screen was added to MODIS C6 and included in VIIRS C1. A QA bit flag is set for all pixels that have a solar zenith angle between 70° < solar zenith angle < 85°. The purpose of this flag is to indicate increased uncertainty when illumination is low.

### 3.2 Clouds

Clouds cause gaps in the snow-cover datasets. Cloud cover can also cause errors related to cloud and snow discrimination in some situations. Discrimination of cloud from snow can be very challenging because some types of clouds and snow can have very similar reflectance characteristics and NDSI values. The current snow cover algorithms use the MODIS cloud mask product (http://modis-atmos.gsfc.nasa.gov/MOD35_L2/index.html) or the VIIRS cloud mask product (JPSS, 2015) to mask clouds. The NASA VIIRS cloud mask product will be used in the snow cover algorithm when it becomes available. The JPSS and NASA cloud mask algorithms are different and since the snow cover algorithm will change to the NASA VIIRS cloud mask product, investigation of cloud and snow confusion in the VIIRS data products will be undertaken at a later date. The NASA VIIRS cloud mask algorithm will be very similar to the MODIS cloud mask thus good consistency between them is expected. A description of cloud masking is given in the MODIS snow products user guide (Riggs et al., 2016a) and the VIIRS snow product user guide (Riggs et al., 2016b). Consistent cloud masking is a significant challenge as



the cloud masking algorithms can be affected by changes in sensor performance and changes in satellite orbit characteristics, and changes in the algorithm, all of which contribute to changes in cloud detection observed between the MODIS C5 and C6, and VIIRS cloud mask products.

## 4 Data products in MODIS Collection 6 (C6)

The MODIS C6 snow-cover products are produced as a sequence of products from the L2 swath product generated with the snow-cover algorithm described above, to L3 gridded and composited daily, eight day and monthly products as shown in Table 1. Snow detection is done in the L2 snow-cover product, which is then used as input to the higher level product where compositing and gridding algorithms are used to generate the products. However, since the M*D10A1 is the most frequently used product, a brief description of that product is given. The higher level products are produced from compositing using

one of the snow-cover products as input. A description of all the products and algorithms can be found in Riggs et al., (2016a).

The M*D10A1 daily gridded snow-cover product is produced from the daily Level 2G M*D10G product. This is an intermediate algorithm and product generated by the MODIS Adaptive Processing System (MODAPS) algorithms to map

M*D10_L2 swath products to the sinusoidal projection of gridded tiles. Each tile is about 10° latitude x 10° longitude with global coverage. All of the swath observations for a day are mapped into the product. The M*D10A1 algorithm selects the 'best' observation of the day, with 'best' defined as the observation nearest local solar noon, closest to nadir and with greatest coverage in a grid cell. That 'best' observation for all datasets is stored in the M*D10A1 product, and subsequently used by the higher level products. In addition to those datasets, there are datasets of pointers to the acquisition time of the

input observations that can be used determine the swath start date and time of an observation mapped into a grid cell (Riggs et al., 2016a). The M*D10A1 product is the most used product based on DAAC distribution metrics and published literature. The MOD10A1 SCE map covering part of central Great Plains and parts of the Rocky Mountains of North America on the last day of 2016 is shown in Fig. 8.

All the M*D10A1 tiles for a day are mapped and composited to the climate modeling grid (CMG) at 5 km spatial resolution to generate the daily global snow cover product, M*D10C1. The MOD10C1 snow cover for 14 January 2017 is shown in Fig. 9. Description of the M*D10A1 algorithm is given in Riggs et al. (2016a).



### 4.1 Major changes in MODIS Collection 6 compared to MODIS Collection 5

#### 4.1.1 NDSI Snow Cover dataset

The MODIS C5 products contained binary SCE and FSC maps, but those datasets have been replaced by the NDSI_Snow_Cover dataset in C6. The binary SCE was derived based on thresholding the NDSI value in the following way:

if the NDSI $\geq 0.4$ then a pixel was mapped as "snow," if not then "not snow." Within that algorithm there was also a snow decision algorithm that combined the NDSI $< 0.4$ and normalized difference vegetation index (NDVI) to improve snow detection in forests (Klein et al., 1998). The FSC was estimated based on empirical regression relationship using the NDSI (Riggs et al., 2006). However use of a threshold value does not optimize the information from the NDSI to detect a large range snow-cover conditions, and the empirical FSC relationship did not allow users to make the best use of the NDSI.

Therefore the binary SCE and FSC maps of C5 and earlier collections have been abandoned and replaced by the NDSI_Snow_Cover and NDSI datasets in C6. An example of the datasets in the M*D10_L2 C6 product is shown in Fig. 10. NDSI is calculated and reported for all daylight land and inland water pixels. The cloud mask is not applied to the NDSI dataset.

The advantage of the NDSI datasets is that a user can generate SCE or FSC maps for their specific study area using their criteria or relationships to derive more accurate maps than could be derived by using one NDSI threshold for the entire globe (as done in C5 with the NSDI $\geq 0.4$ threshold for mapping snow in a pixel). However, if a user wants to estimate FSC using the C5 MODIS regression equation, perhaps to maintain continuity with results derived using C5 data, then the C5 FSC equation can be applied using the NDSI data and masks in the NDSI_Snow_Cover datasets.

#### 4.1.2 The Quantitative Image Restoration (QIR) technique

The Terra and Aqua MODIS instruments are very similar in design and performance, except for Aqua MODIS band 6, in which ~ 75% of the detectors are non-functional (MCST, 2017). Therefore Aqua MODIS band 6 cannot be used directly in the snow algorithm, even though it is an integral part of the MODIS Terra snow algorithm (Eq. 1). In the C5 MYD10_L2

algorithm MODIS band 7 (2.12 μm) was used in place of band 6 and empirically adjusted for band differences, thus the snow detection algorithms for Terra and Aqua were different. The Aqua snow product in C5 was considered less accurate in mapping snow cover as compared to MOD10_L2 because of its use of band 7 instead of the preferred band 6.

For C6, the quantitative image restoration (QIR) algorithm (Gladkova et al., 2012) was applied to restore the missing Aqua

MODIS band 6 data to scientifically-usable data for snow detection. A description of the QIR can be found at http://csdirs.ccny.cuny.edu/csdirs/projects/multi-band-statistical-restoration-aqua. When the MODIS snow algorithm was tested in C5 using the QIR band 6 data, it was found that the output maps were accurate as compared to Terra FSC maps.

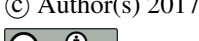



Therefore the QIR algorithm was integrated into C6 processing. Thus for C6 the same snow detection algorithm is run for both Terra and Aqua MODIS instruments using band 6 data and snow-mapping results are now comparable according to our testing.

## 5 Data Products in NASA VIIRS Collection 1 (C1)

The NASA VIIRS snow-cover data products are listed in Table 2. Snow-cover data products are produced in sequence beginning with a swath at a nominal pixel spatial resolution of 375 m. Data product levels are as follows. Level 1B (L1B) is a swath (scene) of VIIRS data geolocated to latitude and longitude. A Level 2 (L2) product (VPN10) is a geophysical product that remains in latitude and longitude orientation of L1B. A Level 2 gridded (L2G) product is in a gridded format of the sinusoidal projection for VIIRS land products. At L2G the data products are referred to as tiles, each tile being 10° x 10°

area, of the global map projection. L2 data products are gridded into L2G tiles by mapping the L2 pixels into cells of a tile in the map projection grid. The L2G algorithm creates a gridded product necessary for the Level 3 (L3) products. An L3 product is a geophysical product that has been temporally and or spatially manipulated, and is in a gridded map projection format and comes as a tile of the global grid. The VIIRS L3 snow products are in either the sinusoidal projection (VNP10A1) or geographic projection (VNP10C1).

An example of the snow cover datasets in the VPN10 product are shown in Fig. 11. Naming of the NASA VIIRS snow cover data products uses a convention similar to the MODIS products. Striping on the sides of the VIIRS datasets in Fig. 11 is the bowtie trim, which is the result of trimming scan lines that overlap during on board data processing (Wolfe et al., 2013).

The VIIRS snow-cover detection algorithms and data products are very similar to the MODIS algorithms and products. Differences are due to sensor spectral band and spatial resolution differences. The higher 375 m spatial resolution of VIIRS results in improved snow-cover mapping along edges snow-covered regions and in snow-covered forests where higher spatial resolution may decrease mixed pixels effects. Initial evaluations of VIIRS snow cover compared to MODIS 500 m snow-cover maps found a slight increase of 1-3% in snow-cover extent on heterogeneous landscapes and along edges of

snow-covered regions. Also reduction of subpixel cloud contamination is possible at the higher resolution. The VIIRS cloud mask product based on the NOAA cloud mask algorithm is used to mask clouds until the NASA VIIRS cloud mask product becomes available for use. Both VIIRS cloud mask products are at 750 m resolution which may have some spatial effects compared to the MODIS cloud mask product at 1 km resolution. The NASA VIIRS cloud mask algorithm/product is expected to be similar to the MODIS cloud mask so when it becomes available, experience gained with the MODIS cloud

mask should be transferable to it. The MODIS and VIIRS SCE products contain the same datasets though the product formats are different. HDF-EOS is used for the MODIS data products and NetCDF4.2/HDF5 is used for the VIIRS L2 data products, and HDF5-EOS is used for L3 products.



## 6 Data Availability

The MODIS snow-cover data products are produced in the MODAPS at the NASA GSFC (https://modaps.modaps.eosdis.nasa.gov/services/) and archived at the NASA NSIDC DAAC. The MODIS C6 data products are available for the entire missions of Terra and Aqua missions. The MODIS C5 ends on 31 March 2017 and data from that

collection will be available until the end of 2017. The NASA VIIRS C1 data products are being produced in the Land Science Investigator-Led Processing System (LSIPS) at NASA GSFC (https://viirsland.gsfc.nasa.gov/index.html) and archived at the NASA NSIDC DAAC. The C1 data products will be available beginning about May 2017. Updates on data collections and release dates are posted by NSIDC. The NSIDC archives and distributes these products free of charge to the user (nsidc.org/daac/). DOIs of the referenced datasets:

MODIS Collection 5

doi: http://dx.doi.org/10.5067/ACYTYZB9BEOS

doi: http://dx.doi.org/10.5067/R90VAMI75N22

doi: http://dx.doi.org/10.5067/63NQASRDPDB0

doi: http://dx.doi.org/10.5067/ZFAEMQGSR4XD

doi: http://dx.doi.org/10.5067/EI5HGLM2NNHN

doi: http://dx.doi.org/10.5067/EW53FPU9NAS6

MODIS Collection 6

doi: http://dx.doi.org/10.5067/MODIS/MOD10_L2.006

doi: http://dx.doi.org/10.5067/MODIS/MYD10_L2.006

doi: http://dx.doi.org/10.5067/MODIS/MOD10A1.006

doi: http://dx.doi.org/10.5067/MODIS/MYD10A1.006

doi: http://dx.doi.org/10.5067/MODIS/MOD10C1.006

doi: http://dx.doi.org/10.5067/MODIS/MYD10C1.006

VIIRS Collection 1

doi:10.5067/VIIRS/VNP10.001

## 7 Conclusions

In this paper we highlighted key changes that have resulted in snow-cover detection improvements in the algorithms and to increase data content in the products. A key improvement in the MODIS C6 algorithm is the way in which the temperature

screen is applied to eliminate errors of snow cover omission in Northern Hemisphere mountains especially during the spring and summer. Another notable improvement is the use of the QIR algorithm applied to restore data from non-functioning detectors in Aqua MODIS band 6, which is a key band for calculation of NDSI in the basic snow-mapping algorithm. Those

changes, along with changes that relate to QA and bit flag reporting, have been implemented to improve MODIS and NASA VIIRS snow-cover algorithms for global, automated SCE mapping from space, while minimizing errors of omission and commission. Data content of the MODIS C6 / VIIRS C1 snow-cover products include the NDSI_Snow_Cover dataset which is the snow-cover map, and additional data layers that include the NDSI data, basic QA value and QA algorithm bit

flags data. Increased data content allows flexibility in use of the data for research and applications. Details of the MODIS C6 / NASA VIIRS C1 and earlier MODIS collections are described in the MODIS and VIIRS Snow Products User Guides (Riggs et al., 2016a & b).

The MODIS and VIIRS snow-mapping algorithms and data products were designed to be very similar to enable the 16+ year

SCE record of MODIS to be extended into the future with S-NPP VIIRS and VIIRS follow-on instruments, thus enabling data continuity and ultimately a climate-data record (CDR) to be created.

**Author contribution**

G. Riggs and D.K. Hall are co-investigator and primary investigators, respectively, on the NASA MODIS and VIIRS Land Science Teams who have developed the snow cover algorithms and data products and documented them since the beginning

of the missions. M. Román coordinates the maintenance and refinement of the MODIS and VIIRS snow cover products. Input to manuscript: all.

**Competing interests**

The authors declare that they have no conflict of interest.

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

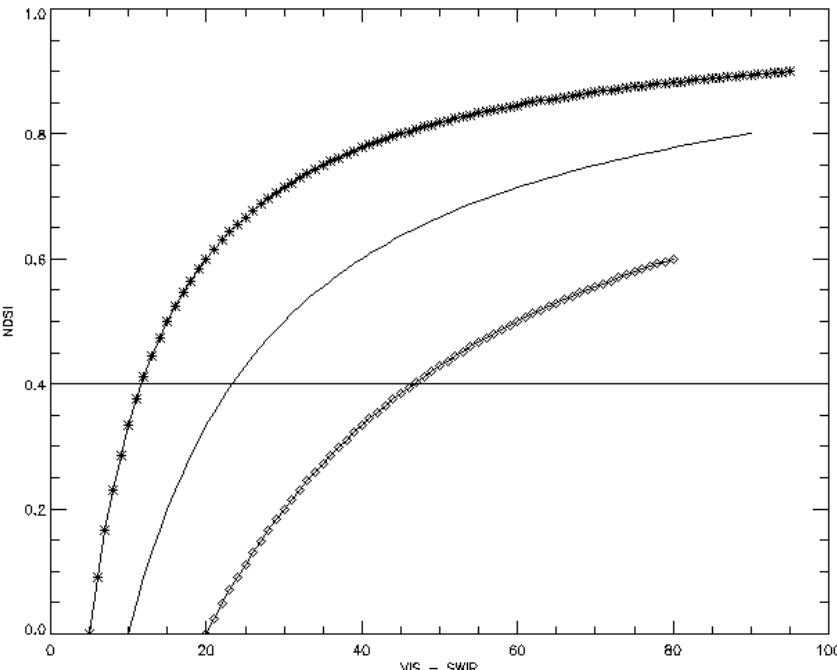

5    **Figure 1. NDSI plots. NDSI values are plotted for three specified conditions. 1) The solid line is the NDSI with the VIS reflectance increasing from 0 to 100% in increments of 1% while the SWIR reflectance is constant at 10%. 2) The line with asterisks shows VIS reflectance increasing from 0 to 100% in increments of 1% while the SWIR reflectance is constant at 5%. 3) The line with diamonds shows VIS reflectance increasing from 0 to 100% in increments of 1% while the SWIR reflectance is constant at 20%.**




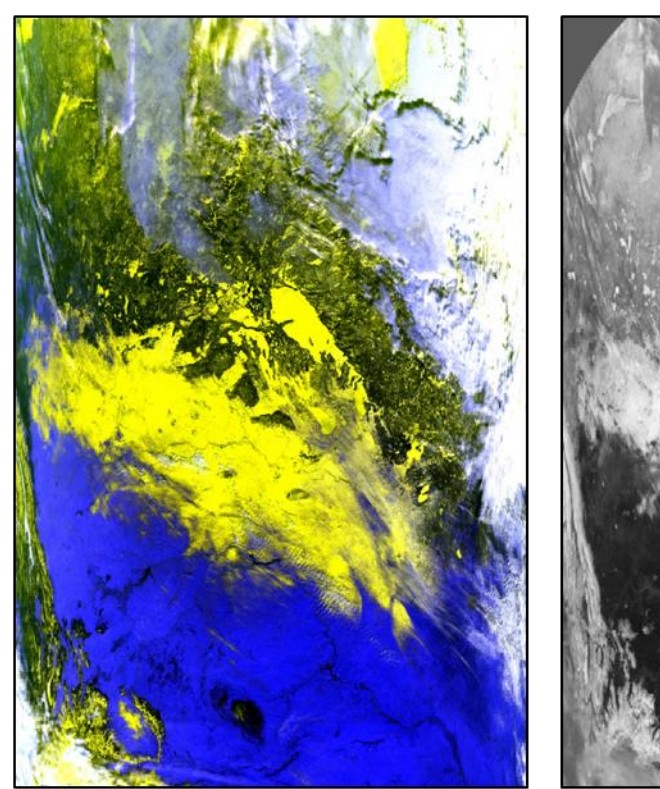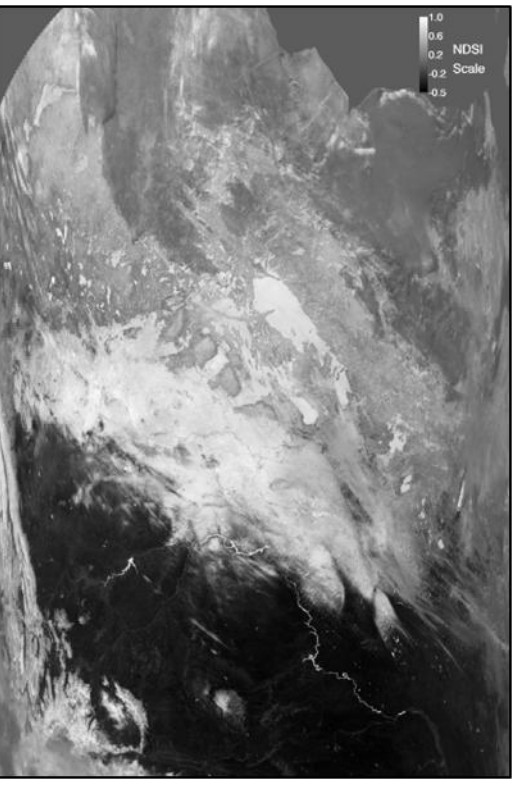

**Figure 2. MODIS NDSI. Swath of MOD10_L2 NDSI dataset (right) for 10 January 2003 1750 UTC compared to false color display of MOD02HKM bands 1,4,6 (left) with snow covered landscape in hues of yellow. NDSI data in range of -1.0 to 1.0, clouds are not masked,while oceans and night are masked. Swath covers central Canada, Lake Winnipeg near center of image, and Plains of the USA. Snow covered forest, snow covered plains and snow free plains are seen in this swath.**



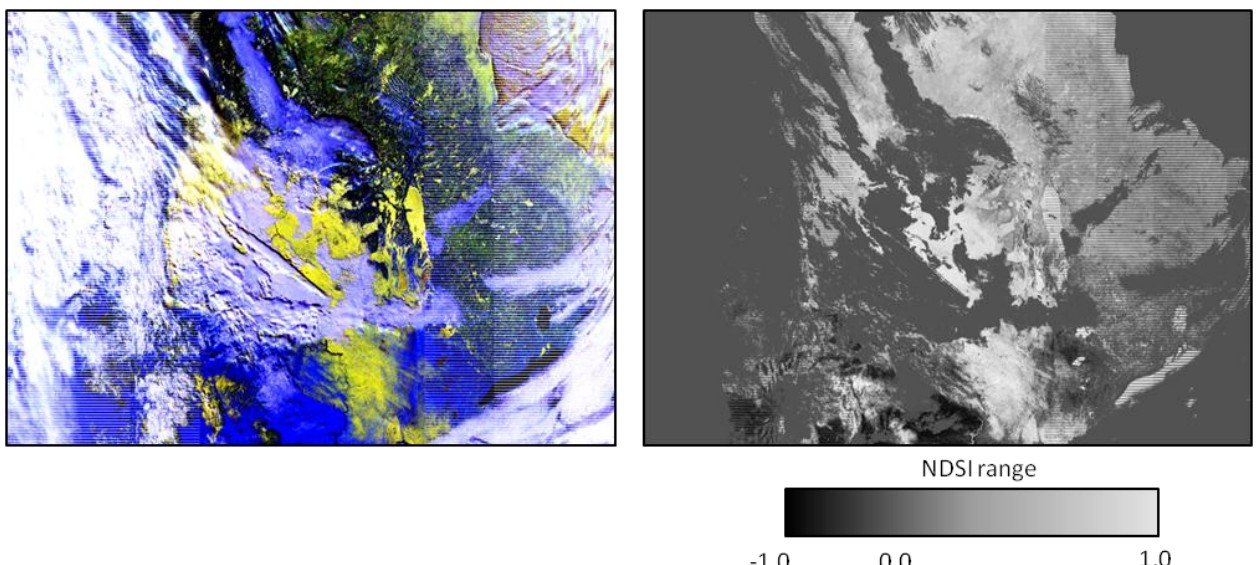

**Figure 3. VIIRS NDSI. VIIRS swath of NDSI dataset (right) for 5 December 2014 1945 UTC compared to false color display of bands I1,I2,I3 (left) with snow covered landscape in hues of yellow. NDSI data in range of -1.0 to 1.0, clouds,oceans and night are masked. Swath covers central Canada, Lake Winnipeg is near the center of the swath and Plains of the USA. Note that the striping and the edge of the VIIRS swath is the result of "pixel trimming" resulting in a reduction of the VIIRS "bow-tie" effect in the NASA Level 1 product (Wolfe et al., 2013).**

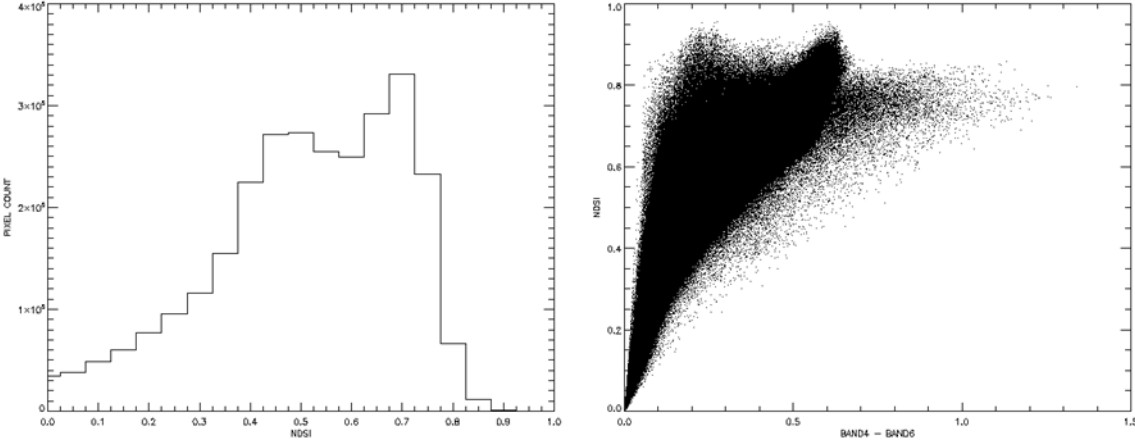

**Figure 4. NDSI histogram (left) and scatter plot of NDSI versus VIS-SWIR difference (right) for the MOD10_L2.A2003010.1750.006 image in Fig. 2. Pixels restricted to snow only, i.e. NDSI > 0.0 and not masked as cloud in the NDSI_Snow_Cover dataset. Histogram was calculated with NDSI bin size of 0.05.**





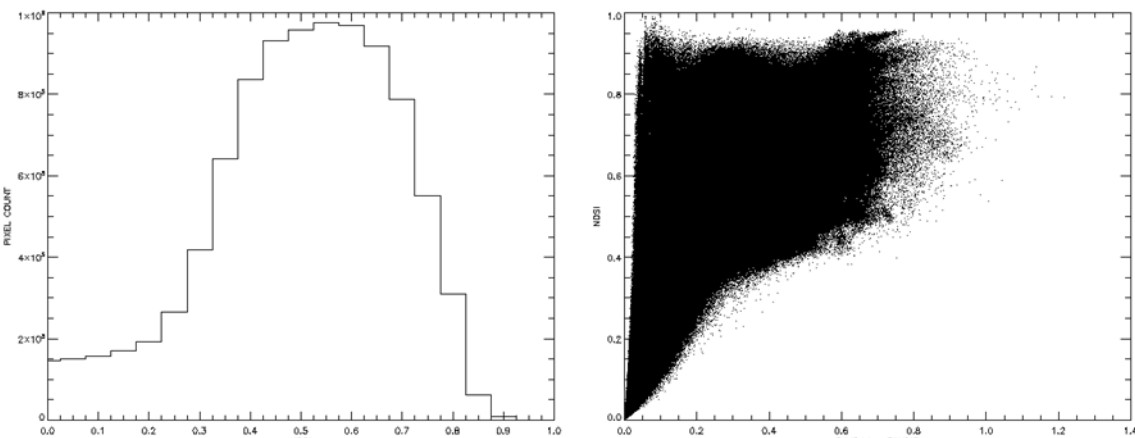

**Figure 5. NDSI histogram (left) and scatter plot of NDSI versus VIS-SWIR difference (right) for the VIIRS 5 December 2015, 1945 UTC image in Fig. 3. Pixels restricted to snow only, i.e. NDSI > 0.0 and not masked as cloud in the NDSI_Snow_Cover dataset. Histogram was calculated with NDSI bin size of 0.05.**



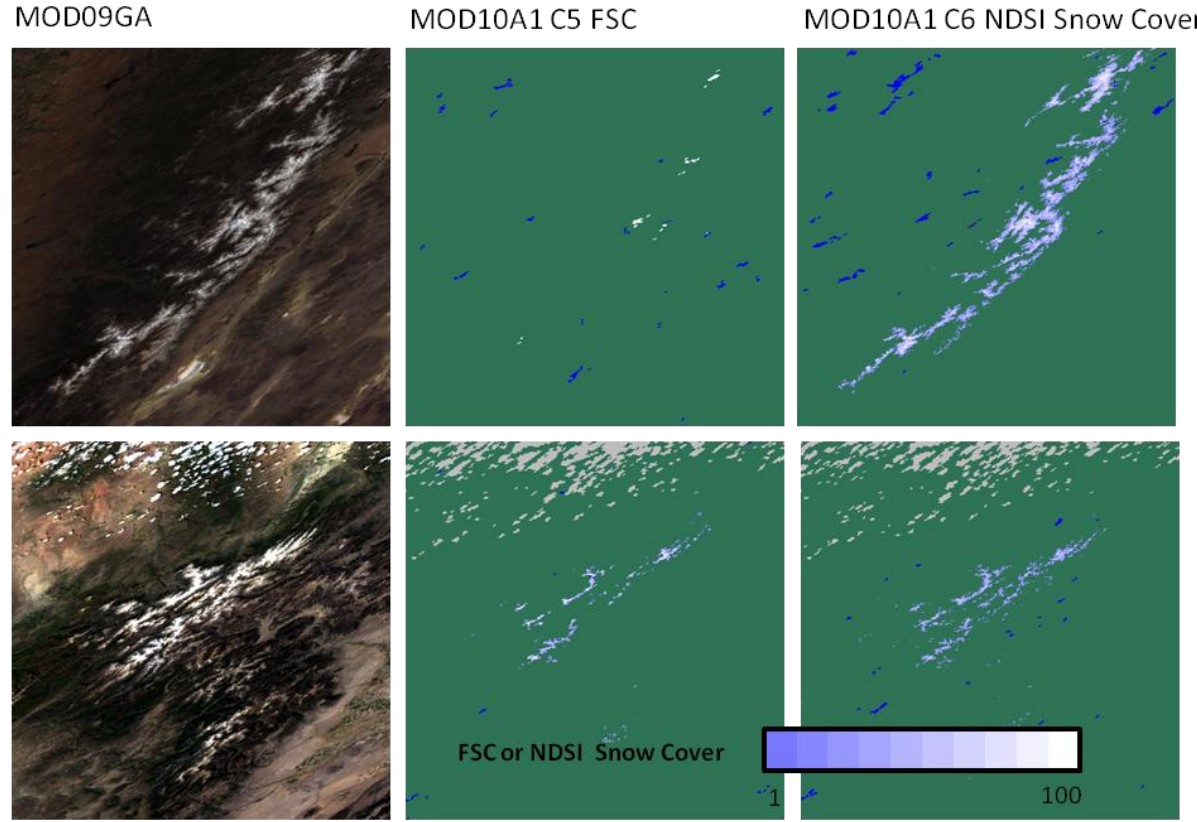

**Figure 6. Comparison of snow-cover extent between MOD10A1 C5 and MOD10A1 C6. Sierra Nevada range on 15 June 2002, top row, and Upper Rio Grande region on 22 May 2002, bottom row. True color (bands 1, 4, 3) image of the MOD09GA is shown in left column. Significant snow cover extent was missed in C5 FSC however all the snow cover extent is detected in C6 NDSI_Snow_Cover by the revised algorithm.**

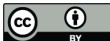



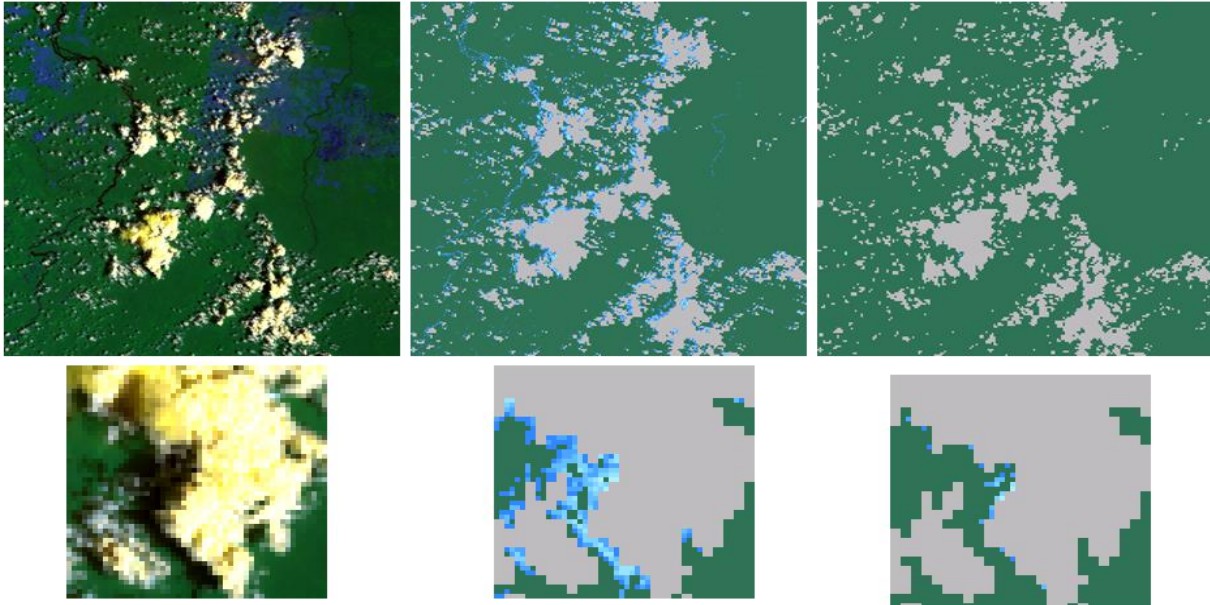

**Figure 7.** Image of southeastern Brazil from VIIRS swath 2014255.1720.PI_03001. False color image of I1, I2, I3 in top left image, zoom of cloud shadow detail in bottom left image. VIIRS NDSI snow cover output without the surface temperature screen in middle images. Top and bottom zoom show the snow commission errors associated with cloud shadowed land and fringes of cloud. VIIRS NDSI snow-cover output with the surface temperature screen applied in right images. Top image and bottom zoom show the effectiveness of alleviating snow commission errors. A few pixels of snow commission error remain at the fringe of clouds, which have NDSI > 0.0, and are below the surface temperature threshold and are not reversed by other screens.

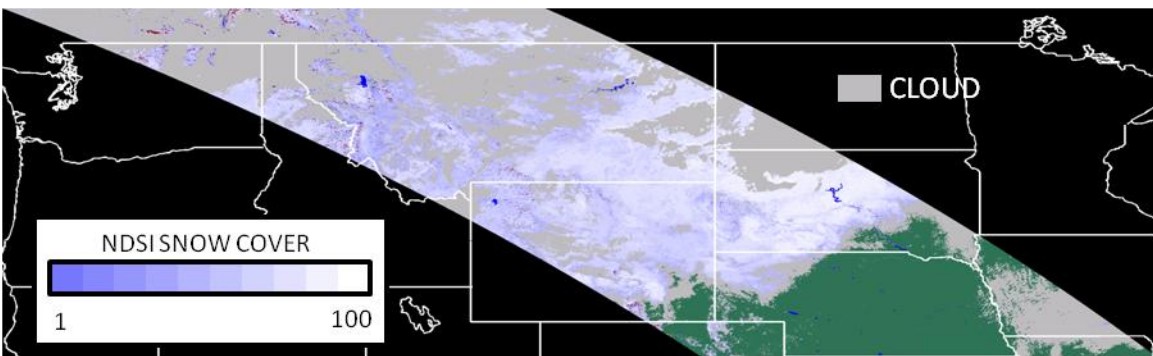

**Figure 8.** Snow cover MOD10A1.A2016366.h10v04.006.*.hdf. Snow cover across the Plains and Rockies on 31 December 2016. The tile has been projected to geographic grid.



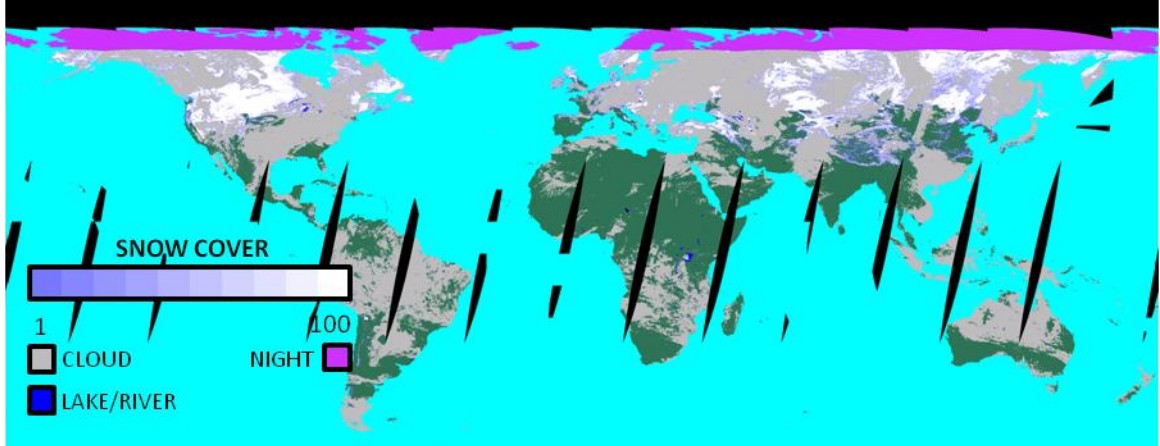

**Figure 9. MODIS global snow cover on 14 January 2017. Global snow cover is from MOD10C1.A2017014.006.\*.hdf. Snow cover is shown with the cloud cover data layer overlaid; the snow cover and cloud cover are in separate data layers in the product.**





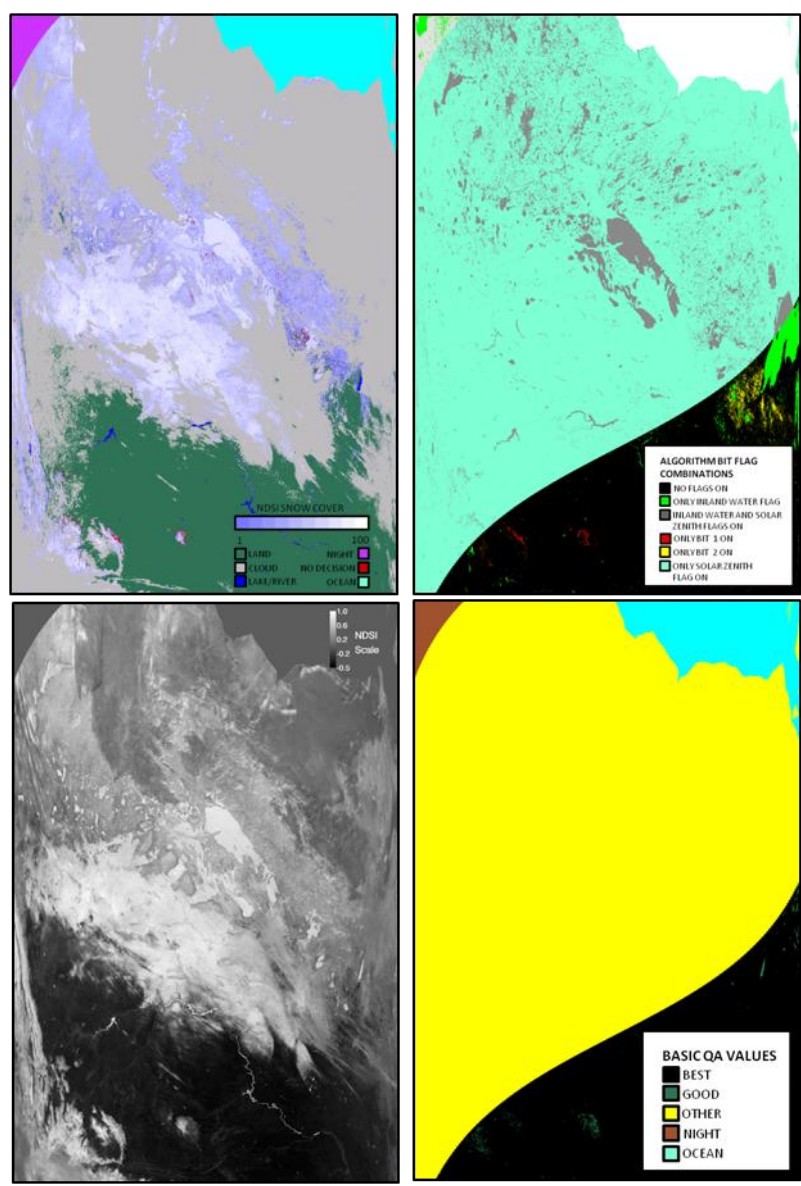

**Figure 10. MODIS C6 MOD10_L2.A2003010.1750, imaging the Great Plains and boreal forest in North America. datasets: NDSI_Snow_Cover (upper left) NDSI (lower left), a selected combination of the Algorithm_bit_flag QA is shown in upper right, and the basic QA is shown in lower right.**





**Figure 11.** VIIRS C1 VNP10 acquired 18 December 2016 at 202 UTC, imaging the western CONUS. Datasets: NDSI_Snow_Cover (A) NDSI (B), Basic QA (C) and a selected combination of the Algorithm_bit_flag QA (D). Unprojected image. The extent of snow cover seen is similar to snow cover extent found on the IMS United States snow cover map (www.ncdc.noaa.gov /snow-and-ice/snow-cover/us/20161218).



**Table 1.  MODIS Terra and Aqua snow data products, Terra (MOD) and Aqua (MYD) products are indicted by M*D.**

| Earth Science Data Type (ESDT) | Product Level | Nominal data Array Dimensions | Spatial Resolution | Temporal Resolution | Map Projection |
|---|---|---|---|---|---|
| M*D10_L2 | L2 | 1354x2030 km | 500 m | 5 min swath | None, lat and lon referenced |
| M*D10GA | L2G | 1200x1200 km | 500 m | daily | Sinusoidal |
| M*D10A1 | L3 | 1200x1200 km | 500 m | daily | Sinusoidal |
| M*D10C1 | L3 | 360°x180°, global | 0.05° x 0.05° | daily | Geographic |
| M*D10A2 | | 1200x1200 km | 500 m | daily | Sinusoidal |
| M*D10C2 | L3 | 360°x180°, global | 0.05° x 0.05° | 8-days | Sinusoidal |
| M*D10CM | L3 | 360°x180°, global | 0.05° x 0.05° | monthly | Geographic |

**Table 2. Summary of land snow-cover products produced at the Land Science Investigator-led Processing System (LSIPS) described at https://earthdata.nasa.gov/about/science_investigator-led-processing-systems/sips-snn--land.**

| Product | ESDT | Description |
|---|---|---|
| Snow Cover L2 Swath | VNP10 | VIIRS/NPP Snow Cover 6-Min Swath 375 m |
| Snow Cover L3 Daily Tile | VNP10A1 | VIIRS/NPP Snow Cover Map Daily L3 Global 375 m SIN Grid Day |
| Snow Cover L3 CMG | VNP10C1 | VIIRS/NPP Daily Snow Cover L3 Global 0.05° X 0.05° climate-modeling grid (CMG) |