# Peer review of "Overview of NASA's MODIS and VIIRS Snow-Cover Earth System Data Records"

_Earth System Science Data, 2017_

## Referee Comment (RC1) · J. Parajka (Referee) · 22 Apr 2017

General comments

This study reviews and describes the development and advancements of MODIS snow cover data records and recent VIIRS snow cover product.

This is a very interesting and clearly written contribution, which is worth to publish. It documents the history and development of the widely used MODIS snow cover datasets and their recent alternative. It contains the most important references related to the methodological development but also studies related to its implementation and testing. I propose only a few minor comments to be eventually considered for revision:

1) Please consider to formulate the objectives somewhat more precisely. The aim

just to describe the snow products is rather general and does not indicate clearly the content of the manuscript.

2) Figures 4,5: The axis labels are very difficult to read

3) Figure 7: Some legend is missing here. Is Brazil a relevant region for comparison?

---

## Referee Comment (RC2) · Anonymous Referee #2 · 1 May 2017

This is a very good and concise overview of MODIS and VIIRS snow retrieval algorithms and of the snow products. Overall the paper looks fine to me and it is practically ready for being published. It would be great however if the authors could address the following few questions and comments in the final version of the manuscript. (1) As I understand the snow thermal screen is turned off at 1300m surface elevation and above. Is there any justification for choosing this particular value ? (2) Although cloud shadows are mentioned in the paper it is not clear whether they are somehow accounted for in the algorithm or in the product. Are snow retrievals performed in shadowed areas ? If they are, the quality of retrievals should probably degrade. It would be good if you could also comment on the possible effect of topographic shadowing on MODIS and VIIRS snow products. (3) It is stated that the VIIRS snow detection algorithm is very similar to MODIS. Could you please elaborate on this issue a bit further ? Are the al-

gorithms indeed identical or they have differences ? Apparently the central wavelength of VIIRS I1 is not the same as MODIS band 4 (see page 6). Does this difference have any noticeable effect on NDSI ?

———————————————————

---

## Referee Comment (RC3) · Anonymous Referee #3 · 9 May 2017

The manuscript presents an overview of the latest MODIS and VIIRS dataset which is very important for cold region study. This dataset is absolutely unique and significantly useful since the previous versions of MODIS snow cover dataset has been widely used over the world. I look forward to use this new dataset and hope this manuscript can be published as early as possible. Only two technical suggestions: 1) In the abstract, line 12, an abbreviation 'Suomi-NPP' is used but it is changed to 'S-NPP' in line 26. 2) In page 8, line 15-20, the authors discussed 'it was common in spring and early summer on some mountain ranges for mixed pixels that included snow to have surface temperatures above 283K and thus a great deal of spring or summer snow on mountains was not mapped'. But in line 22-25 ,the authors say similar sentences again :'They showed that snow cover in the seasons of autumn, spring and summer on some mountain ranges was not mapped in the MOD10A1 C5 FSC data product'. I suggest the section

3.1.1 can be reorganized for better understanding for readers.

---

## Referee Comment (RC4) · Anonymous Referee #4 · 10 May 2017

The manuscript provides overview of recent modification of two essential for snow-cover investigation Earth System Data Records: MODIS and VIIRS. The content and scope of executed work is great, informative and, certainty, is worth to be published within a very short time. Based on read text, I propose only a few general and more specific corrections which can be taken into account at the will of author. First of all, for me it was difficult to perceive a structure of manuscript clearly. The reason, probably, lies in not enough explicit image of average reader. The task to combine a few Guide books and wide literature review to produce short overview is very difficult, that is why often the authors refer to different sources foe additional reading. Therefore, the purpose of the overview is to provide only recent improvements and new achievements, and as it was mentioned in manuscript in the beginning,it should be clear along whole article just like the fact that the average reader is already familiar with previous data

sets. That is why when on page 10 a brief description of daily product appears due to, as it pointed, its popularity, I think, It overloads text and confuses the reader concerning a purpose of the paper (if it is written to cover popular points or new ones). Another purpose of overview is to be more understandable especially to compare with reading of guide books. For that matter, I found this text a little bit too technical. Sometimes, it is difficult to read, and a lack of coherence is felt. Some example is presented here (page 6, line 16-18): "Snow cover has an NDSI > 0.0. However other features e.g. salt pans, and cloud contaminated pixels at the edges of cloud, and features with very low visible reflectance can have NDSI > 0.0, and thus be erroneously detected as snow, which results in a snow commission error". Probably, the word "too" after second mentioning of NDSI threshold is missed, and it leads to difficulties in perception. Another structural confusion consists in providing reasons of C5 algorithm improvement only by the end in 4th chapter. I find that they should be delineated in the beginning, but at the same time, I see the difficulties to put it before theoretical backgrounds. Therefore, the main advice is to revise a text and structure one more time to make it more plain and coherent. More specific comments concern, at first, already mentioned by one of referee, 1300 threshold for mountains. It would be really interesting to read possible explanation of such chosen value and why it is "linked with height"(page 8, line 28). Another point which needs an explanation, as I see it, the choice of data format (page 12, line 31-32). As it is placed in text, it is interesting to know why this difference takes place. Further on, on the figure 6, C5 FSC and C6 NDSI Snow Cover are compared. The advantage of C6 is visible here but it is interesting to compare them with C5 Snow cover data too. Actually, the question about whether this improvements so significant or not appears a few times along text. As "significantly" improved product on line 12 of page 3 turns into "notable" (line 29 on 7 page) while "the accuracy of the MODIS C6 products is similar to, or better than MODIS C5". At the same time, the merits of new data set is still obvious, but a lack of more critical improvement assessment confuses a little bit in a few moments of different chapters.

In conclusion, I want to admit that a huge work had been done and such an article will

be very useful for every researchers in sphere of snow investigation. I hope that my notes will help to make the text with description of this important work a little bit more clear and structured.

---

## Author Comment (AC1) · 26 May 2017

Juraj, we appreciate your comments on the manuscript. We will revise based on your comments. 1) We will work on clarifying the objective of the manuscript. 2) The text on the graphs will be improved. 3) The example over Brazil (Fig. 7) was used because it's a good example of the cloud/snow confusion situation. However that location is irrelevant for monitoring snow cover, so will be replaced with an example from a boreal region in spring or summer season. This type of cloud/snow confusion is most clearly seen in a season when the surface is snow free.

---

## Author Comment (AC2) · 30 May 2017

The data content of the VIIRS snow cover Level 2 data product VNP10 described in Section 5 and at doi:10.5067/VIIRS/VNP10.001 in the discussion paper has been revised since the initial posting of the paper. No change in the VIIRS snow cover detection algorithm or the snow cover data sets in VNP10 has been made. The VNP10 has been revised to include latitude and longitude coordinate datasets and relevant dataset attributes to enable netCDF4 CF georeference of the product by software tools that read netCDF4 data format. The revision in data product content and format was prompted by recommendations that came from recent meetings of Earth science data product and information system groups producing and archiving the data products. Release of the VNP10 product has been delayed to July 2017 because of the revisions and required testing of the product for archiving at the NSIDC DAAC.